# Discovery of Three Toxic Proteins of *Klebsiella* Phage fHe-Kpn01

**DOI:** 10.3390/v12050544

**Published:** 2020-05-15

**Authors:** Cindy M. Spruit, Anu Wicklund, Xing Wan, Mikael Skurnik, Maria I. Pajunen

**Affiliations:** 1Department of Bacteriology and Immunology, Medicum, Human Microbiome Research Program, Faculty of Medicine, University of Helsinki, 00290 Helsinki, Finland; c.m.spruit@uu.nl (C.M.S.); anumaria.wicklund@gmail.com (A.W.); xing.wan@helsinki.fi (X.W.); mikael.skurnik@helsinki.fi (M.S.); 2Laboratory of Microbiology, Wageningen University and Research, 6708 WE Wageningen, The Netherlands; 3Division of Clinical Microbiology, HUSLAB, University of Helsinki and Helsinki University Hospital, 00290 Helsinki, Finland; 4Department of Microbiology, Faculty of Agriculture and Forestry, University of Helsinki, 00790 Helsinki, Finland

**Keywords:** bacteriophage, *Podoviridae*, *Drulisvirus*, hypothetical proteins of unknown function, toxic proteins, antibiotic resistance, antibiotics, *Klebsiella pneumoniae*, capsule type

## Abstract

The lytic phage, fHe-Kpn01 was isolated from sewage water using an extended-spectrum beta-lactamase-producing strain of *Klebsiella pneumoniae* as a host. The genome is 43,329 bp in size and contains direct terminal repeats of 222 bp. The genome contains 56 predicted genes, of which proteomics analysis detected 29 different proteins in purified phage particles. Comparison of fHe-Kpn01 to other phages, both morphologically and genetically, indicated that the phage belongs to the family *Podoviridae* and genus *Drulisvirus.* Because fHe-Kpn01 is strictly lytic and does not carry any known resistance or virulence genes, it is suitable for phage therapy. It has, however, a narrow host range since it infected only three of the 72 tested *K. pneumoniae* strains, two of which were of capsule type KL62. After annotation of the predicted genes based on the similarity to genes of known function and proteomics results on the virion-associated proteins, 22 gene products remained annotated as hypothetical proteins of unknown function (HPUF). These fHe-Kpn01 HPUFs were screened for their toxicity in *Escherichia coli*. Three of the HPUFs, encoded by the genes *g10*, *g22,* and *g38,* were confirmed to be toxic.

## 1. Introduction

Infections with antibiotic-resistant bacteria cause 700,000 deaths per year and the number of fatalities is predicted to drastically increase to 10 million per year by 2050 [1]. *Klebsiella pneumoniae* is a rod-shaped bacterium that is responsible for one-third of the infections caused by gram-negative bacteria. *K. pneumoniae* isolates constitute a major source of antibiotic resistance that has steadily increased over the years [2]. Infections with extended-spectrum antibiotic-resistant *K. pneumoniae* cause prolonged hospitalization, high mortality, and high economic burden [3]. Antibiotic resistance is acquired by both accumulating mutations and horizontal gene transfer (HGT) of antibiotic resistance between bacteria. HGT is the major cause of the development of extremely drug-resistant bacteria [2]. The lack of treatment for infections with multidrug-resistant (MDR) gram-negative bacteria is considered as the biggest unmet medical need [4]. The World Health Organization listed the carbapenem-resistant and extended-spectrum beta-lactamase (ESBL)-producing *Enterobacteriaceae*, which include among others *K. pneumoniae* and *Escherichia coli*, as the critical pathogens for which new antibiotics should be developed [5]. Treating infections of MDR bacteria with phage therapy may be one of the treatment options, although most phages have a very narrow host range, which makes finding the right bacteriophage a challenge [6,7]. The development of new antibiotics is therefore still believed to be the most promising solution to tackle resistant bacteria [4].

Promising novel antibiotics may be developed from phage proteins. Phage endolysins, virion-associated peptidoglycan hydrolases, polysaccharide depolymerases, and holins are already extensively studied for their use as antibiotics. Phages use these enzymes to infect or lyse bacteria [8]. Many more bacteriophage proteins specifically interact with bacterial proteins to ensure an efficient infection cycle. Interestingly, the functional mechanisms of these interactions are largely unknown [9]. Furthermore, 70% of the predicted genes in phage genomes are annotated to encode for hypothetical proteins of unknown function (HPUFs) [10,11,12]. The HPUFs mostly encode small proteins ranging from 30 to 200 amino acids. The great potential of using HPUFs as toxic proteins has already been shown [13,14,15,16,17]. The HPUFs may be toxic via yet unknown mechanisms and have the potential to be developed into innovative antimicrobial peptides.

Here, we describe the isolation and characterization of phage fHe-Kpn01 that infects an ESBL strain of *K. pneumoniae*. A systematic screen of the HPUFs present in the genome of fHe-Kpn01 revealed the presence of three proteins that are toxic to *E. coli*, providing new leads for developing innovative antibiotics.

## 2. Materials and Methods

### 2.1. Bacterial Strains, Phage, and Media

*E. coli* laboratory strain DH10B [18] was used for routine plasmid DNA isolation and cloning. *E. coli* DH5α [19] was used in the experiment to confirm protein toxicity. Electrocompetent cells for *E. coli* were prepared, stored, and used as described previously [20]. The *K. pneumoniae* strains used in this work are described in Appendix A and were mostly obtained from the hospital district of Helsinki and Uusimaa laboratories (HUSLAB), Finland. Media (agar, tryptone, tryptone soy broth, yeast extract) were purchased from Neogen Food Safety, Lansing, MI, USA. Unless indicated otherwise, bacterial and phage incubations were performed at 37 °C using Luria Broth (LB) [21]. Soft agar LB medium included additionally 0.4% (*w*/*v*) agar and LB agar plates were solidified with 1.5% (*w*/*v*) of agar. M9t minimal medium contained 3.4 mM Na_2_HPO_4_, 2.2 mM KH_2_PO_4_, 0.86 mM NaCl, 0.94 mM NH_4_Cl, 0.2% (*w*/*v*) tryptone, 2.0 mM MgSO_4_, 0.10 mM CaCl_2_ and 3.0 × 10^−3^ mM vitamin B1 (Merck KGaA, Darmstadt, Germany). Bacterial strains were stored at −80 °C in 3% (*w*/*v*) tryptone soy broth with 20% (*v*/*v*) glycerol. Phage fHe-Kpn01 was stored at −80 °C in LB supplemented with 8% dimethyl sulfoxide.

### 2.2. Phage Isolation and Purification

fHe-Kpn01 was isolated from a municipal sewage sample collected in Helsinki, Finland, using clinical ESBL *K. pneumoniae* strain #5504 (Appendix A) as the host. The same strain was used as the standard host strain for the phage. Phage lysates were produced from semiconfluent soft-agar plates as described elsewhere [21]. The fHe-Kpn01 lysate was ultrafiltrated with Amicon Ultra-4 (100 kDa) centrifugal filter units (Merck KGaA, Darmstadt, Germany) to one-quarter of the initial volume. Three volumes of SM buffer (100 mM NaCl, 50 mM Tris pH 7.5, 10 mM Mg_2_SO_4_, 0.01% (*w*/*v*) gelatin (Type B from bovine skin, Merck KGaA, Darmstadt, Germany)) were added and ultrafiltration was repeated. The phages were purified further by discontinuous glycerol density gradient ultracentrifugation at 40,000 rpm at 4 °C for 4 h in a Beckmann Coulter SW55Ti swing-out rotor. The phages were resuspended after ultracentrifugation in SM buffer containing 8% (*w*/*v*) sucrose. Purified phage particles and intermediate purification products were stored at 4 °C.

### 2.3. Electron Microscopy

Ultra-centrifuged phage particles were sedimented by centrifugation at 16,000× *g* at 4 °C for 2 h and resuspended in 0.1 M ammonium acetate, pH 7.2. Subsequently, the phage particles were allowed to sediment on 200 mesh Formvar-coated copper grids for 1 min and stained negatively using 1% uranyl acetate pH 4.2 (method modified from [22]). Stained particles were observed using a JEOL JEM1400 electron microscope operated at 80 kV at the Department of Virology, University of Helsinki. Pictures were taken using an Olympus Morada CCD camera.

### 2.4. Host Range Determination

The host range of fHe-Kpn01 was determined by pipetting droplets of 10 μL of serial dilutions of concentrated phage stocks on lawns of different bacterial strains prepared on LB agar plates. 72 *K. pneumoniae* strains, one *K. oxytoca*, and one *E. coli* strain (negative control) were used (Appendix A). The plating efficiency, measured by the formation of plaques of lysis, was tested at 37 °C. Positive droplet test results were confirmed by the double-layer method using appropriately diluted phage preparations.

### 2.5. Infection Growth Curves

An overnight bacterial culture of *K. pneumoniae* #5504 was diluted 100-fold in fresh LB medium and 270-µL aliquots were distributed into Bioscreen Honeycomb 2 plates wells (Growth Curves Ab Ltd., Helsinki, Finland), where they were mixed with 30 µL of different fHe-Kpn01 phage stock dilutions. The phage stock and bacterial culture were mixed to achieve multiplicity of infection (MOI) values ranging between 10^−6^ and 10^−2^. The control consisted of 270 µL of bacterial culture and 30 µL of fresh LB medium. The growth experiment was carried out at 37 °C using a Bioscreen C incubator (Growth Curves Ab Ltd., Helsinki, Finland) with continuous shaking. The optical density at 600 nm (OD_600_) of the cultures was measured every hour for up to 15 h. The averages were calculated from values obtained for the bacteria grown in five parallel wells.

An overnight bacterial culture of *K. pneumoniae* #6326 was diluted to a ratio of 1:10 in fresh LB medium, and 180-µL aliquots were distributed into Bioscreen Honeycomb 2 plates wells, where they were mixed with 20-µL aliquots of different fHe-Kpn01 phage stock dilutions. The phage stock and bacterial culture were mixed to achieve MOI values ranging between 10^−6^ and 10^−2^. The control consisted of 180 µL of bacterial culture and 20 µL of fresh LB medium. The growth experiment was carried out at 37 °C using a Bioscreen C incubator with continuous shaking. The OD_600_ of the cultures was measured every 45 min for up to 15 h. The average values were calculated from values obtained for the bacteria grown in four parallel wells.

### 2.6. Genome Sequencing and Analysis

Phage DNA was obtained from high-titer phage preparations as described earlier [21] or using the Invisorb Spin Virus DNA Mini Kit (Stratec Biomedical, Birkenfeld, Germany). Sequencing was performed at Eurofins Genomics. The next-generation sequencing DNA library (insert size of 665 ± 370 nucleotides) was paired-end sequenced using Illumina MiSeq sequencer (Illumina, San Diego, CA, USA) with a read length of 150 nucleotides. The A5 (Andrew And Aaron’s Awesome Assembly)-miseq integrated pipeline for de novo assembly of microbial genomes was used to assemble the genome sequence [23]. The termini of the phage genome were identified using PhageTerm [24], restriction digestions and Sanger sequencing using phage DNA as template and the *fliC* fragment ligation approach [25]. The orientation of the genome was arranged similarly to the sequences of closely related homologs as found in a nucleotide BLAST search. The genes were annotated with RAST software [26] and validated manually confirming also that the predicted genes were accompanied by a properly located ribosomal binding site.

A protein BLAST against the non-redundant protein sequences database (release 2.9.0 from April 1, 2019) was performed for every predicted gene product and the two results with the lowest E-values were recorded (Appendix A). Furthermore, every gene product was analyzed using HHpred [27] and the best hits with a probability above 50% and an E-value below 1 were recorded (Appendix A). The presence of tRNAs was investigated using tRNAscan-SE [28]. In addition, Res-Finder-3.1 [29] and VirulenceFinder-2.0 [30] software was used. The protein content, as tryptic peptides, of purified phages was analyzed using liquid chromatography-tandem mass spectrometry (LC-MS/MS) at the Proteomics Unit, Institute of Biotechnology, University of Helsinki as described earlier [17,31]. Calibrated tryptic peptide peaks were searched against the predicted tryptic peptides from the amino acid sequences of all open reading frames (ORFs), even the non-annotated ones, in the genome of fHe-Kpn01. The proteins identified by LC-MS/MS analysis as having two or more unique tryptic peptides and over 5% sequence coverage were annotated as phage (structural) proteins. The complete genome sequence with annotation was deposited in the NCBI nucleotide database (GenBank) under accession number MN380459.

Bacterial genomic DNA from *K. pneumoniae* strains #5504 and #6326 was isolated from a 5-mL overnight culture using the NucleoSpin Microbial DNA Kit (Macherey-Nagel, GmbH Düren, Germany) following the manufacturer’s instructions. Sequencing was performed at Novogene Europe (Cambridge, United Kingdom) using Illumina HiSeq with 150-bp paired-end reads. The genomic scaffolds of the strains were assembled de novo from the raw Illumina reads with SPAdes genome assembler v3.14.0 [32]. The genome sequences of *K. pneumoniae* strains ATCC 10031, ATCC 43816, and ATCC 700721 were retrieved from the ATCC Genomes database. Multi-locus sequence types and the K (capsule) and O antigen (lipopolysaccharide, LPS) serotypes of the *K. pneumoniae* strains were predicted using Kleborate tool [33,34]. The raw sequence read data of *K. pneumoniae* strains #5504 and #6326 was deposited to the NCBI sequence read archive under bioproject PRJNA627626.

### 2.7. Restriction Fragment Analysis of Phage DNA

Restriction digestions of purified fHe-Kpn01 DNA were performed with restriction endonucleases *Bam*HI, *Box*I, *Cla*I, *Not*I, *Nsi*I (Thermo Fisher Scientific, USA), *Acc65*I, *Dra*I, *Msc*I, and *Sal*I-HF (New England Biolabs, Ipswich, MA, USA) in appropriate digestion buffers.

### 2.8. Detection of Nicks in the Phage DNA

To detect nicks in the genome of fHe-Kpn01, approximately 300 ng of purified fHe-Kpn01 genomic DNA was incubated for 30 min at RT with 200 U S1 nuclease (Thermo Fisher Scientific, USA) in the supplied reaction buffer.

### 2.9. Phylogenetic Analysis

A nucleotide BLAST search of the complete genome of fHe-Kpn01 against the nucleotide collection (nr/nt) database with optimized conditions for highly similar sequences was used to detect the phages that are most closely related to fHe-Kpn01. A phylogenetic tree was created using fHe-Kpn01, all drulisviruses present in the ICTV virus taxonomy (release 2018b) [35], and the type species of the other subfamilies present in the family *Autographivirinae* with more than one species per subfamily. The genome-BLAST distance phylogeny (GBDP) method [36] with settings appropriate for prokaryotic viruses [37] was used to conduct pairwise comparisons of the nucleotide sequences. The obtained intergenomic distances were used to create a balanced minimum evolution tree with branch support inferred from 100 pseudo-bootstrap replicates each. The tree was created via FASTME with subtree pruning and regrafting postprocessing [38] using the formula D0. Rooting was performed at the midpoint [39] and the tree was visualized using FigTree [40]. Taxon boundaries at the family, genus, and species level were assessed using the OPTSIL program [41], the favored clustering thresholds [37], and an F value (fraction of links required for cluster fusion) of 0.5 [42].

### 2.10. Initial Screening of HPUFs for Toxicity

Toxic and non-toxic control genes and fHe-Kpn01 genes of HPUFs were cloned into the multiple cloning site of a pUC19-based vector pU11L4 [17] using a three times molar excess of the insert. Cloning was performed using the *Not*I and *Nco*I restriction sites (Thermo Fisher Scientific, USA). Whenever an internal *Nco*I site was present in one of the genes, *Nhe*I (Thermo Fisher Scientific, USA) digestion was used instead (Appendix A). RegB, a restriction endoribonuclease from phage T4 [43] was used as the toxic control. As non-toxic control, the structural phage capsid vertex protein gene product 178 (Gp178) from phage φR1-RT [17,44] was used. Electrocompetent DH10B *E. coli* cells with a transformation efficiency between 10^8^ and 10^9^ colony forming units (CFUs) per µg of intact pU11L4 plasmid (5387 bp in size) were used. Preparation of the electroporation mixtures and melting of electrocompetent cells was performed on ice. Ligation mixtures, containing the equivalent of 10 ng vector backbone, were directly electroporated into 45 µL of electrocompetent DH10B *E. coli.* Electroporation was performed in 2 mm electroporation cuvettes (Bulldog Bio, Portsmouth, NH, USA) and a pulse was given using a Gene Pulser II in combination with a Pulse Controller Plus (Bio-Rad Laboratories, USA) set at 200 Ω, 25 µF and 2.5 kV. Immediately after giving the pulse, the cells were transferred to 950 µl SOC medium (2% (*w*/*v*) tryptone, 0.5% (*w*/*v*) yeast extract, 10 mM NaCl, 2.5 mM KCl, 10 mM MgCl_2_, 10 mM MgSO_4_, 20 mM glucose) and incubated at 35 °C while shaking vigorously. After one hour, an appropriate amount (25–50 μL) of transformed cells was plated in triplicate on LB agar plates supplemented with 100 µg/mL ampicillin (Merck KGaA, Darmstadt, Germany). The plates were incubated overnight at 37 °C and the CFUs per transformation were determined by counting the number of colonies the following day. In general, in each experiment, six fHe-Kpn01 genes were tested simultaneously with the two control genes. The CFUs were standardized to the CFU of the *g178* gene in an individual experiment. Two biological replicates of each fHe-Kpn01 gene were used for electroporation.

### 2.11. Confirmation of Protein Toxicity

Genes of potentially toxic fHe-Kpn01 HPUFs (selected in the initial screening assay) and negative and positive controls *regB* and *g178* were cloned into the pBAD33 vector [45] using *Kpn*I and *Xba*I (Thermo Fisher Scientific, USA). Cloning was performed with *Sph*I when there was an internal *Xba*I restriction site (Thermo Fisher Scientific, USA). After plasmid purification, the presence of the correct inserts was confirmed by sequencing at the Institute for Molecular Medicine Finland Technology Centre Sequencing Unit [46]. Three single colonies per gene (plasmids in DH5α *E. coli*) were inoculated in LB medium (1.5 mL) supplemented with 20 µg/mL chloramphenicol (Clm, Merck KGaA, Darmstadt, Germany) and 0.2% (*w*/*v*) glucose in 15 mL centrifuge tubes. The tubes were incubated overnight at 37 °C while shaking vigorously. Afterward, the bacteria were washed by centrifuging for 5 min (3100× *g*) and replacing the medium with 1.5 mL M9t medium. Then, M9t medium supplemented with 20 µg/mL Clm and either 0.2% (*w*/*v*) glucose or 0.2% (*w*/*v*) arabinose was inoculated with 1% inoculum. The solutions were transferred in triplicate (300 µL/well) to Bioscreen Honeycomb 2 plates. The OD_600_ was measured every hour for 20 h using the Bioscreen C MBR (Oy Growth Curves Ab Ltd, Helsinki, Finland). The plate was shaken continuously with normal speed and high amplitude, shaking was stopped 5 sec before measuring, and measuring was started immediately. The overall mean values and standard deviations were calculated over the three biological replicates.

### 2.12. Prediction of the Structure of Toxic Proteins

The structures of the toxic proteins were modeled using Phyre2 [47].

## 3. Results

### 3.1. General Genomic Features of fHe-Kpn01

Phage fHe-Kpn01 was isolated from a municipal sewage sample collected in Helsinki, Finland. The genome sequence was assembled using 250,000 reads with an average whole-genome coverage of 827 x. The length of the genome is 43,329 bp with a GC-content of 53.6%. Direct terminal repeats of 222 bp are present on both ends of the genome. The genome was digested by at least eight endonucleases (Appendix A), which indicates that the genome consists of unmodified double-stranded DNA. The approximate termini of the genome were determined by restriction enzyme digestion analysis (Appendix A) and comparison of the genome to similar phages (Figure 1). The exact termini were determined using the PhageTerm tool [24] and the *fliC* fragment ligation approach (Appendix A) [25]. The ligation experiments and Sanger sequencing confirmed that the termini are blunt. In the PhageTerm analysis, in addition to the read coverage peaks of the physical termini of the genome, a third major peak at nucleotide position 7873 with high read coverage was detected (Appendix A). The third peak is most likely due to a nick in the genome, since fragments of the expected size were generated when the genome was digested with an S1 nuclease that recognizes single-stranded DNA (Appendix A).

The fHe-Kpn01 genome contains 56 predicted genes that occupy 94.0% of the whole coding capacity. The predicted genes only occur in the forward orientation and encode proteins ranging from 38 to 1232 amino acids in size. An ATG start codon was used for 52 genes, while the remaining four genes started with a TTG start codon. An overview of the predicted proteins, the length of the proteins, and the closest homologs is presented in Appendix A. LC-MS/MS analysis confirmed the presence of 29 proteins in phage particles (Appendix A) which can be considered as phage particle associated proteins (PPAPs) as some of them are clearly structural and some are just carried over to function early in infection. No tRNAs genes were found in the genome. Additionally, no known genes encoding integrases, lysogeny- or virulence-associated proteins, or acquired antimicrobial resistance genes were identified and therefore this bacteriophage can be considered as virulent and potentially safe for phage therapy.

### 3.2. Comparison of fHe-Kpn01 to Closely Related Phages

The genome size (43 kb), size of the icosahedral capsid (59 nm), and the short tail (Appendix A) of fHe-Kpn01 are typical characteristics of a podovirus. Moreover, the closest related phage to fHe-Kpn01 is the *Klebsiella* phage vB_KpnP_SU503, a phage in the family *Podoviridae,* subfamily *Autographivirinae,* and genus *Drulisvirus* [48]. A 96.5% overall nucleotide identity, with 95% query coverage, was found between the phages. Comparison of the genomes of fHe-Kpn01, vB_KpnP_SU503, and KP34 [49], the type species of the drulisviruses, reveals that the genomic organization is very similar in all three phages (Figure 1). Furthermore, a lysis gene cluster consisting consecutively of genes encoding a spanin, a holin, and an endolysin was identified in both fHe-Kpn01 (genes *g54-g56*, Appendix A) and other drulisviruses [48]. Phylogenetic comparison of fHe-Kpn01, nine drulisviruses and six other phages in the family *Autographivirinae* generated the tree as shown in Figure 2, with an average support of 38%. The OPTSIL clustering yielded sixteen species clusters, six genus clusters, and one family cluster. Phage fHe-Kpn01 groups with the other drulisviruses in the phylogenetic tree. We suggest classifying fHe-Kpn01 in the family *Podoviridae*, subfamily *Autographivirinae,* and the genus *Drulisvirus.*

### 3.3. fHe-Kpn01 Host Range and Growth Characteristics

Host range experiments revealed that fHe-Kpn01 has a narrow host range as it was able to infect only three of the 72 tested (mostly) clinical strains of *K. pneumoniae* (Appendix A). The strain used for isolation of the phage (#5504, an ESBL *K. pneumoniae* strain) and strain #6326 [51] were efficiently infected. The phage formed round clear 4-5 mm diameter plaques with haloes on strain #5504 (Appendix A). Strain #5529 is ampicillin-resistant and was only infected by fHe-Kpn01 with very low efficiency. While the capsular serotype of strain #5529 is not known, the capsular serotype of the fHe-Kpn01 sensitive strains #5504 and #6326 was predicted to be KL62 based on the whole genome sequence data (Appendix A). On the other hand, the known capsular serotypes of some of the phage-resistant strains were KL1, KL2, KL49, KL64, or KL107 (Appendix A). We conclude that fHe-Kpn01 should be classified as a KL62 specific phage.

To examine the efficiency of fHe-Kpn01 infection, *K. pneumoniae* strains #5504 and #6326 were infected in liquid culture at different MOI values and the bacterial growth was assessed. The study showed that fHe-Kpn01 can efficiently lyse both cultures at MOIs of at least 10^−6^ (Figure 3). However, re-growth of the bacterial culture was observed within the 15 h period of the experiment, indicating the presence of phage-resistant bacteria. In coherence, the prolonged incubation of bacteria infected with fHe-Kpn01 in soft agar resulted in the emergence of resistant bacteria within two days of incubation, indicating a relatively high rate of phage-resistance development among the bacteria. This phenomenon is in contrast to the low rate of phage-resistance development seen with Twort-like phages [31]. It could be a common feature for podoviruses as we have observed it with some of our yersiniophages or as a common property of *Klebsiella* as observed previously by Tan et al. [52].

### 3.4. Antibacterial Potential of fHe-Kpn01 

On average two-thirds of the predicted gene products of newly sequenced phage genomes are annotated as HPUFs [10,11,12], among which some are found to possess toxic properties [13,14,15,16,17]. Bioinformatic analysis (BLASTP) assigned a predicted function to only 22 out of 56 gene products of fHe-Kpn01, so 34 were annotated as HPUFs. To exclude virion-associated (likely non-toxic) proteins from the HPUFs, LC-MS/MS analysis of purified phage particles was carried out. Among the altogether 29 virion-associated proteins (Appendix A) were 12 HPUFs that were excluded from further toxicity screening. Thus, 22 true HPUFs remained to be included in the toxicity screen (Appendix A).

We systematically screened all HPUFs by cloning each PCR-amplified HPUF-gene into the pU11L4 plasmid. The resulting ligation mixtures as such were electroporated into *E. coli* DH10B. Bacteria were spread onto LB agar plates and the amount of CFUs was counted on the following day. The CFUs relative to the CFU of the non-toxic control Gp178 are displayed in Figure 4 (details in Appendix A). The cloning of the toxic control gene *regB* resulted in a significantly lower number of transformants (6%) than the non-toxic control gene *g178*. Interestingly, among the 22 HPUFs of fHe-Kpn01, the number of transformants varied and five HPUF genes, *g10, g22, g23, g35,* and *g38,* with the lowest ratios relative to *g178* (below 0.4 with a maximum SD of 0.15)*,* were considered to encode potentially toxic proteins.

### 3.5. Three Toxic Proteins are Encoded in the Genome of fHe-Kpn01

The initial screening assay was performed with crude ligation mixtures. Therefore, the numbers of transformants could be affected by the presence of unwanted ligation products, possibly reflected in the relatively high variation between the replicates (Figure 4). To confirm the toxicity of the five selected potentially toxic proteins, their genes were cloned into the pBAD33 plasmid under an arabinose-inducible promoter and transformed into *E. coli* DH5α. When grown in minimal medium supplemented with arabinose, the genes cloned into pBAD33 are expressed, while expression is inhibited in the presence of glucose. As a consequence, the growth curves of bacteria carrying a toxic gene should differ in the presence of either glucose or arabinose, while for bacteria containing a non-toxic gene the growth curves should be identical. Figure 5A shows similar growth curves for all transformed bacteria in the presence of glucose, except for the bacteria containing the toxic control gene *regB*, which grow slightly more slowly. This could be due to partial leakage of the glucose-repression. On the other hand, the growth curves in the presence of arabinose reveal clear toxic effects, in addition to RegB, for Gp10, Gp22, and Gp38, while Gp23 and Gp35 appear to be non-toxic. (Figure 5B). Therefore, we conclude that Gp10, Gp22, and Gp38 of fHe-Kpn01 demonstrate toxicity in *E. coli*.

### 3.6. The Structure of Gp22 Resembles Nucleotidyltransferases

Gp10, Gp22, and Gp38 of fHe-Kpn01 are predicted to be proteins of 69, 182, and 146 amino acids in size, respectively (Appendix A). To get an idea of the possible functions and mechanisms of toxicity of these proteins, the protein sequences were analyzed by HHpred [27] and the structures of the proteins were modeled using Phyre2 [47]. No similar proteins or structures were found for Gp10 by either HHpred or Phyre2. For Gp38, HHpred indicated resemblance with an N-acetyltransferase with a probability of 97.7% and an E-value of 0.0044 (Appendix A), but no structure was predicted by Phyre2. Gp22 was predicted to resemble a nucleotidyltransferase by both Phyre2 and HHpred with over 90% confidence, 73% sequence coverage, and at most 29% amino acid identity by Phyre2 and a probability of 99.8% and E-value of 3.3 × 10^−19^ in HHpred. The closest related structures to that of Gp22 were nucleotidyltransferases of *Aquifex aeolicus* [53,54], human mitochondria [55,56], *Bacillus stearothermophilus* [57], *Thermotoga maritima* [58], *E. coli* [59], and *Planococcus halocryophilus* [60].

## 4. Discussion

We here describe the isolation and characterization of *Klebsiella* phage fHe-Kpn01. The 43,329 bp genome has direct 222 bp terminal repeats and contains 56 predicted genes, all in the forward orientation (Figure 1, Appendix A). The capsid diameter was determined to be 59 nm and the phage has a short tail (Appendix A). The closest homolog to fHe-Kpn01 is *Klebsiella* phage vB_KpnP_SU503 and fHe-Kpn01 is classified in the family *Podoviridae*, subfamily *Autographivirinae,* and the genus *Drulisvirus.* The combined results of the restriction analysis of the genome (Appendix A), Sanger sequencing (Appendix A), comparison of the genome to closely related phages (Figure 2), and PhageTerm analysis [24] (Appendix A) confirmed the location of the physical termini and the 222 bp direct terminal repeats in the genome of fHe-Kpn01. The presence of a third peak in the output of PhageTerm was noticed, besides the two peaks indicating the termini of the genome (Appendix A). An excess of sequencing reads was previously used as an indicator for localized single-stranded interruptions, also known as nicks, in the genome of some podoviruses [61,62] and siphovirus PR1 [63]. The DNA library of fHe-Kpn01, which was used for the sequencing of the phage genome, was prepared by mechanical shearing of the DNA. Nicks in the genome are prone to mechanical shearing and therefore a majority of the DNA molecules could break at this specific location during the preparation of the library. Treatment of the fHe-Kpn01 genomic DNA with S1 nuclease, a nuclease that recognizes single-stranded DNA, indeed indicated the presence of a nick in the genome of fHe-Kpn01, which probably corresponds to nucleotide position 7873 as indicated by PhageTerm (Appendix A). A nick in the genome is also observed in, among others, siphoviruses T5 [64] and BF23 [65], and podoviruses tf [66], φkF77 [67] and LIMElight [68]. Since nicks are present in the genome of several phages, it is important to have a critical look at the output generated by PhageTerm, especially when multiple peaks are present, to prevent an incorrect annotation of the genomic termini.

Initially, fHe-Kpn01 was isolated for usage in phage therapy. Indeed, fHe-Kpn01 is suitable to use in phage therapy since it is a lytic phage and does not encode any known bacterial virulence or antibiotic resistance genes. The host range of fHe-Kpn01 is narrow and therefore the application of this phage in phage therapy would be limited. To elucidate the host specificity of fHe-Kpn01, we sequenced two of the three fHe-Kpn01 sensitive *K. pneumoniae* strains (Appendix A), #5504 and #6326, and based on the Kleborate analysis (Appendix A) both are predicted to be of capsule type KL62 and O-serotype O1v1. Since the capsule is the most common receptor for *Klebsiella* phages, we conclude that fHe-Kpn01 should be classified as a KL62 specific phage. The phage specificity to a bacterial strain depends on the compatibility of the phage tail fiber receptor binding domain and the receptor structure on the cell surface [7]. Therefore, major differences are detected between the sequences of receptor binding proteins (RBPs) in closely related phages with different host ranges, specifically in the receptor binding domains. Two tail components have been classified as RBPs in KP34viruses, corresponding to Gp49 and Gp57 of fHe-Kpn01 [48,69]. The Gp49 RBP homolog appears to occupy a structural role in the tail fiber architecture without any role in the host specificity, leaving that role to the Gp57 homolog [48,69]. Phage vB_KpnP_SU503, the closest relative of fHe-Kpn01, is also very likely KL62-specific as its predicted RBPs, SU503_45 and SU503_53, are 100% and 96.4% identical to Gp49 and Gp57, respectively (Appendix A). Gp57 is a 500 aa protein with only 18 residues that are different from those of SU503_53, with 14 of those scattered in the C-terminal half of the protein (Appendix A). Likely, the N-terminal end of Gp57 contains the pectate lyase enzymatic domain, allowing for degradation of the KL62 capsule [48,69].

Discovering toxic proteins from phages is possible in several ways. First of all, interactions between known bacterial targets and phage proteins can be investigated. Van den Bossche et al. [15] used affinity chromatography to find the interacting phage proteins of nine macromolecular complexes involved in transcription, DNA replication, fatty acid biosynthesis, RNA regulation, energy metabolism, and cell division of *Pseudomonas aeruginosa*. Seven phages (myoviruses, podoviruses, and siphoviruses with genomes ranging from 42 to 72 kb and one genome of 280 kb) were screened and 37 interacting phage proteins were found. Eight toxic proteins were detected, of which four were also toxic in *E. coli* [15]. Although toxic proteins can be discovered using this method, many toxic proteins may be overlooked that interact with other, yet unknown, bacterial targets. To find new bacterial targets of phage proteins, the starting point of the research should be shifted from screening bacterial targets towards screening phage proteins.

One such method makes use of a genomic library of sheared phage genome fragments. Singh et al. [14] generated a genomic library of seven siphoviruses with a genome of 49 to 57 kb in which the genome fragments were expressed from an inducible promoter. The clones were screened for their toxicity in *Mycobacterium smegmatis* and *Mycobacterium bovis*. Two toxic phage proteins, that were also toxic in *E. coli*, and two toxic synthetic peptides, that were encoded by artificial ORFs, were identified [14]. The genomic fragments that are randomly cloned into plasmids may generate artificial ORFs, with original fragments in the wrong orientation, or contain a truncated ORF. Furthermore, some ORFs in longer inserts may not be expressed from the inducible promoter that is present in the plasmid. As a result, there is a high chance of overlooking toxic phage proteins. A more thorough screening assay would investigate phage ORFs individually.

Liu et al. [13] cloned all predicted genes, except for those encoding holins, amidases, and proteins with predicted transmembrane domains, of 27 *Staphylococcus* phages individually into plasmids under the expression of an inducible promoter. The toxicity of the phage proteins was screened in *Staphylococcus*. In one of the phages, a siphovirus with a genome of approximately 42 kb, three toxic proteins were detected. In all 27 phages together (964 ORFs), 31 novel toxic protein families were found. Highly related proteins, which belong to the same family, were found in several phages [13]. We believe that the method used by Liu et al. [13] is very suitable for detecting all toxic proteins encoded by phages. There may, however, be a slightly more efficient way of screening all genes of phages that does not require the purification of plasmids and uses methods to exclude certain genes from the screening assay.

In the approach used previously by our group, we excluded both gene products with a predicted function and virion-associated (structural) proteins from the screen, since both types of proteins are unlikely to be toxic via novel mechanisms [17]. The selected HPUF genes of the 169 kb myovirus φR1-RT were cloned into plasmids and the ligation mixtures were used as such in the initial screening assay to transform *E. coli* bacteria. The toxicity of proteins in the initial assay, recorded as a lower plating efficiency, was confirmed in a follow-up assay in which the candidate genes were cloned into the plasmid pBAD30 under the expression of an inducible promoter. Four toxic proteins were found in the genome of this phage [17]. Characterization of the toxic mechanisms of those proteins is undergoing.

In the present work, the HPUFs of fHe-Kpn01 were screened following the above-described strategy [17] with some modifications in the initial screening assay to obtain a more robust assay. During the optimization of the initial screening assay, we found that variation occurs in several stages of the assay (Figure 4 and Appendix A). First of all, many pipetting steps are used during the assay. Secondly, the assay utilizes electroporation, which is rarely used during a quantitative assay. Moreover, although the use of unpurified ligation mixtures saves time, it may also introduce some unwanted plasmids into the bacteria, causing variation between ligation mixtures of the same gene. Furthermore, only limited amounts of electrocompetent cells could be made in one batch and therefore several batches of cells with slightly different transformation efficiencies were used. However, by using *E. coli* as the primary screening host, high quality cells could be obtained for the electroporation of crude ligation mixtures with comparable efficiencies. In addition, variation could occur in the spreading and growth of the bacteria on plates, possibly resulting in variation in CFU counts even between replicates from a single transformation experiment. Lastly, only a limited amount of electroporations could be performed at the same time to ensure equal treatment of all tested genes. Generally, we performed eight electroporations at the same time, which included six HPUFs, the toxic control, and the non-toxic control. The amount of CFUs of the non-toxic control Gp178 seemed to be the most stable factor during all electroporations. Therefore, the CFU numbers were normalized relative to the CFU numbers of Gp178 in each batch of electroporations. Despite the different sources of variation during the initial screening assay, we obtained reliable results by using two biological replicates and three plates per electroporation. The five HPUF proteins showing the lowest relative transformation efficiencies (Figure 4) were tested in a second assay that confirmed the toxicity of three of the proteins (Figure 5). Considering the studies that have been performed previously and the genome sizes of the phages screened in those studies, it is likely that we found most of the toxic proteins of fHe-Kpn01.

The discovery of completely new toxic proteins can provide novel insights into bacterial targets and mechanisms of toxicity. The first step in investigating the discovered toxic proteins in more detail was carried out in silico by investigating whether structural domains of the proteins could be predicted. While no structural elements could be predicted for Gp10 and Gp38, the structure of Gp22 was found to be related to tRNA nucleotidyltransferases, also called 3’-terminal CCA-adding enzymes, of several species. Nucleotidyltransferases are part of a big superfamily of proteins that stretches over archaea, bacteria, and eukaryotes. Proteins in this superfamily contain a common core fold and several conserved catalytic residues, but often little sequence similarity is found between different members of this superfamily [70]. The proteins belonging to this family are involved in nearly all cellular processes, including transcription, translation, regulation of protein activity, DNA repair, RNA editing, cell signaling, and cell death [71,72]. Common sequence patterns are found in distinct members of this superfamily, including tRNA nucleotidyltransferases [71]. Interestingly, these sequence patterns are also found in the sequence of Gp22, starting at amino acid 29 (hG[GS]), 44 ([DE]h[DE]h) and 114 (h[DE]h), in which h indicates a hydrophobic amino acid. The latter two patterns contain the catalytic residues [71] and are located in the highly conserved beta-sheet region of these types of proteins [73]. All in all, it is possible that Gp22 functions as a nucleotidyltransferase. The mechanism by which Gp22 exerts its toxicity is, however, unclear since these proteins are involved in many different cellular processes.

The next step in the exploitation of the toxic character of Gp10, Gp22, and Gp38 requires in-depth characterization of their mechanisms of toxicity. Investigating the toxicity of the proteins in both gram-negative and -positive bacteria could indicate whether conserved host protein complexes are targeted [14]. To examine the toxicity in different host cells, one should consider using shuttle vectors or, alternatively, Gateway vectors for trivial cloning steps. We have used *E. coli* as a screening host for toxic proteins of *Yersinia* [17] and *Klebsiella* (in this work) phages, reasoning that the hits would cross the species- and genus-barriers thus making the new targets more universal. Identification of the protein-protein-interaction networks of the toxic proteins would require the production and purification of affinity-tagged proteins. This could be challenging in bacteria as the proteins are toxic. However, an in vitro transcription-translation approach could be used. Subsequently, the proteins could be structurally and biophysically characterized and their bacterial targets could be elucidated. Smaller peptides with better solubility could be synthesized from the toxic proteins to investigate their antibacterial properties. In addition, the bacterial targets could be used to screen synthetic compound libraries for small molecules that can mimic the effect of the phage proteins [13]. Ultimately, the discovery of the three toxic proteins of fHe-Kpn01 could lead to the development of new antibiotics.

In summary, we isolated and characterized the novel *Klebsiella* phage fHe-Kpn01. By systematically screening the HPUFs encoded in the genome, we identified three toxic proteins, which opens up opportunities for the development of innovative antibiotics.

## Figures and Tables

**Figure 1 viruses-12-00544-f001:**
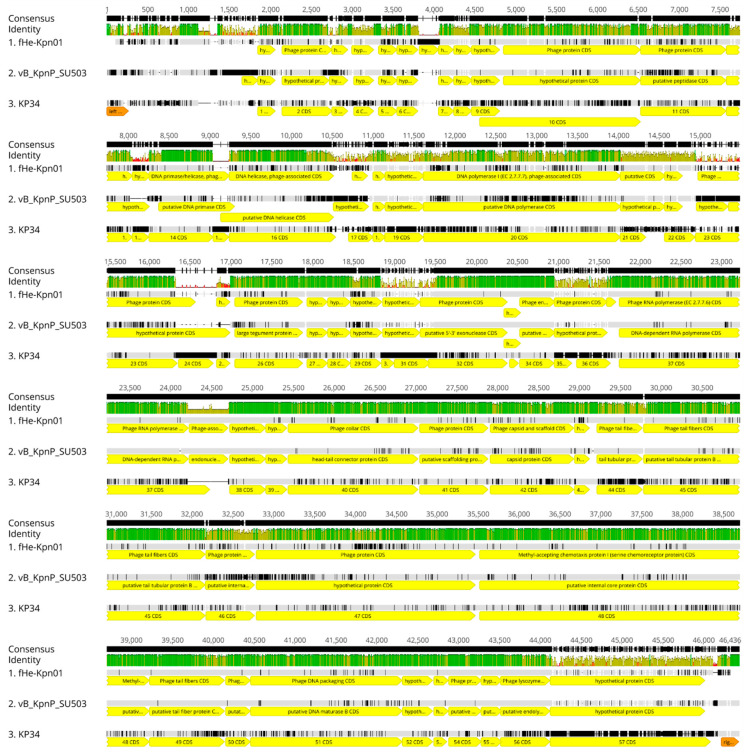
Alignment of the fHe-Kpn01 genome with the closest related phage vB_KpnP_SU503 (NC_028816; 43,809 bp) and KP34 (NC_013649: 43,809 bp), the type phage of the drulisviruses. The genes are represented by yellow arrows. A green area in the consensus identity indicates high similarity and a red area indicates low similarity between the three phages. Generated with Geneious v11.1.5 [50].

**Figure 2 viruses-12-00544-f002:**
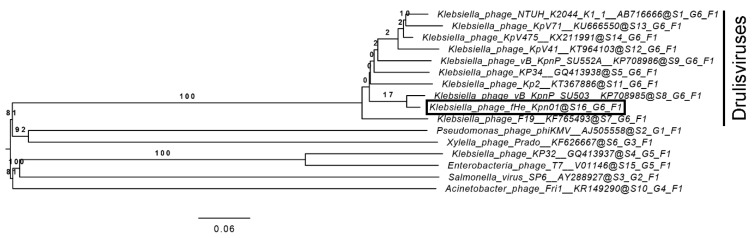
The phylogenetic genome-BLAST distance phylogeny (GBDP) tree of fHe-Kpn01, the drulisviruses present in the ICTV virus taxonomy (release 2018b) [35], and the type species of other subfamilies present in the family *Autographivirinae*. The numbers above branches are GBDP pseudo-bootstrap support values (100 replications). The branch lengths are scaled in terms of the used formula D0.

**Figure 3 viruses-12-00544-f003:**
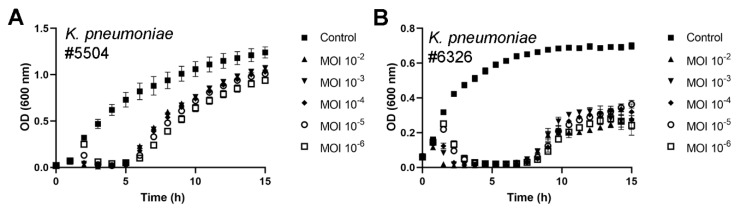
Growth curves of *Klebsiella pneumoniae* strains #5504 (**A**) and #6326 (**B**) infected with fHe-Kpn01. Bacteria were cultured with different initial multiplicities of infections (MOIs) in liquid LB-medium at 37 °C. Each data point represents the average optical density (OD) at 600 nm for four to five replicates and error bars represent the standard deviation (SD).

**Figure 4 viruses-12-00544-f004:**
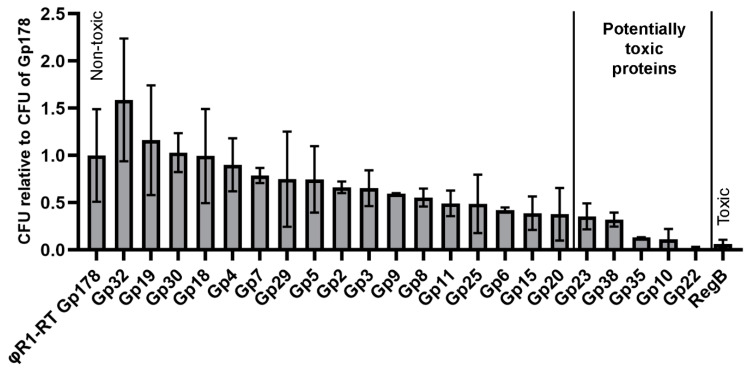
Toxic gene screening by transformation efficiency plating showing the average number of colony forming units (CFU) of transformants (± SD of two biological replicates) relative to that of *g178*. The ligation mixtures of the pU11L4 backbone with the hypothetical proteins of unknown function (HPUF) genes of fHe-Kpn01 or the control genes *regB* (toxic, a restriction endoribonuclease gene from phage T4) or *g178* (non-toxic) were electroporated into *E. coli* DH10B. For each ligation mixture, the number of transformants was determined from triplicate plating. The cut-off value for potentially toxic gene products (Gp) was set to 0.4 with maximum SD of 0.15.

**Figure 5 viruses-12-00544-f005:**
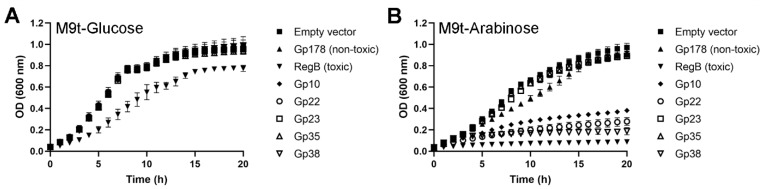
Growth curves of *E. coli* DH5α expressing the indicated gene products (Gp) under the arabinose-inducible promoter of pBAD33 in M9t minimal medium supplemented with repressing glucose (**A**) or inducing arabinose (**B**).

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
