# Peer review of "Discovery of Three Toxic Proteins of Klebsiella Phage fHe-Kpn01"

_viruses, 2020, doi:10.3390/v12050544_

Round 1
Reviewer 1 Report
The authors present an interesting and comprehensive study on the bacteriophage fHe-Kpn01 that infects an extended-spectrum beta-lactamase-producing strain of Klebsiella pneumoniae. The obtained results are important for understanding of diversity of this group of bacteriophages as well as evolution of Podoviridae. In addition, authors show that several viral proteins are toxic for Escherichia coli cells. Overall, the methods used to study the bacteriophage are appropriate and the results are presented in a clear way, however a few minor issues should be addressed to accept the manuscript for publishing.
Minor issues, comments, suggestions
Lines 24, 385 and several more. Authors declare that “any harmful genes” exist in the genome of the phage or “any toxins” are encoded by phage. However, they identified the toxic proteins in the same genome. Intuitively, it is clear what authors are discussing, but the more strict definitions should be included into the text to discriminate those different “toxicities”. Moreover, nobody knows which of the viral genes would be toxic if it is synthesized inside the eukaryotic cell. Hence, we should be very aware of using such generalizations.
Another issue is related to the research objectives. It is not fully clear why E. coli is used as a target for testing of the protein toxicity. Is that due to a simple system or different reasons take place? This point needs a more detailed justification. Authors also suggest that the discovered proteins might be new leads for developing innovative antibiotics. However, it is not fully clear how that may happen, keeping in mind that all three proteins are intracellular in contrast to hydrolases or holins, which act from the outside of the cell. An appropriate discussion would be welcome, especially keeping in mind that E. coli itself can produce a lot of toxic proteins, e.g. toxins from toxin-antitoxin systems, which, following the authors’s ideas, can be also considered as innovative antibiotics.
Several technical questions/suggestions.
The determination of the chromosome ends as well as a nick is based only on sequencing data. An analysis by a primer extention reaction would be suggested , too. Moreover, the figure presenting the terminal sequences (are the ends blunt, 5prim- or 3prim-protruded?) as well as sequences around a nick should be included.
The authors used Phyre2 for the prediction of structures of the hypothetical proteins and failed to find any function for Gp38. However, a HHpred analysis of Gp38 shows that this protein might be a representative of the N-acetylases. Hence, it is strongly recommended to re-check the putative functions of all annotated ORFs by HHpred and amend the appropriate table.
Author Response
Please see attachment. Same also below:
The authors fully agree with the first comment and the text (lines 24, 254 and 395; Note that all line numbers are now with track changes on and show all modifications) is now modified accordingly to be more specific.
We chose E .coli as the primary host for screening as the transformation efficiencies are much higher and more repeatable compared to Klebsiella species. That does not diminish the importance of using the original host for further studies (lines 500-503). We added one sentence to clarify this aspect (lines 463-464).
The authors realize that there is still a huge step(s) between our quest to search potentially toxic phage proteins to new antibacterial drugs and thus we have added a short discussion on that topic (lines 506-510).
The determination of the physical ends of the genome is now described in more detail (in the methods (line 134), results (line 239) and discussion (line 379) and illustrated in Figure S2. The direct terminal repeat as well as the sequence around the predicted nick are shown in Figures S2 and S3, respectively.
We added the HHpred analysis for all ORFs in Table S2, where we recorded the proteins with the highest probability (whenever the probability was above 50% and the E-value was below 1, as recommended by HHpred). The methods (paragraph 2.6, starting at line 126) and results (paragraph 3.6, starting at line 357) were adjusted accordingly.

Reviewer 2 Report
The manuscript describes the isolation and characterization of a new Klebsiella bacteriophage, which belongs to genus Drulisvirus family Podoviridae and the identification of potentially toxic proteins, encoded by the phage genome. The manuscript is thoroughly written, all methods are clear and reproducible, the discussion is comprehensive. Several comments related to the presentation of results are offered.
- The image in figure 1 is not of good quality, it is hard to see the structure of the bacteriophage particle. Please replace it with a better image.
- Figure 2: It is not clear what genome do the genes marked with yellow arrows correspond to. The function of most genes is also unclear since their names are incomplete. At least the main proteins encoded by the genome should be signed.
- Fig. 5: Please clarify, what was the criteria to select putative toxic proteins? Why did you choose gp23 and gp38? Transformation efficiency plating did not differ significantly for constructions, containing genes Gp6, Gp15, Gp20, Gp23, and Gp38.
Author Response
Please see the attachment. The same also below:
Unfortunately we do not have a better image of the phage particle (old Figure 1), thus we moved the image to the Supplementary file (Figure S4) as we thought that in the current situation and with respect to the main message of the manuscript, getting a better image is not necessarily relevant.
Figure 1 (the multiple alignment; note new numbering) is now more detailed.
In our previous study (Mohanraj et al, 2019) the cut-off value was set to 0.4 (percentage to non-toxic control), which we now specified also to take into account SD values at maximum 0.15. We clarified the criteria in the text (line 326 and 332, Figure 5 legend).
